



# North Atlantic Ocean–Atmosphere Driven Variations in Aerosol Evolution along Lagrangian Cold-Air Outbreak Trajectories

Kevin J. Sanchez[1,2], Bo Zhang[3], Hongyu Liu[3], Matthew D. Brown[2,4], Ewan C. Crosbie[2,4], Francesca Gallo[1,2], Johnathan W. Hair[2], Chris A. Hostetler[2], Carolyn E. Jordan[2,3], Claire E. Robinson[2,4], Amy Jo Scarino[2,4], Taylor J. Shingler[2], Michael A. Shook[2], Kenneth L. Thornhill[2,4], Elizabeth B. Wiggins[1,2], Edward L. Winstead[2,4], Luke D. Ziemba[2], Georges Saliba[5], Savannah L. Lewis[5], Lynn M. Russell[5], Patricia K. Quinn[6], Timothy S. Bates[6,7], Jack Porter[9], Thomas G. Bell[8,9], Peter Gaube[10], Eric S. Saltzman[9], Michael J. Behrenfeld[11], and Richard H. Moore[2]

[1]NASA Postdoctoral Program, Universities Space Research Association, Columbia, MD
[2]NASA Langley Research Center, Hampton, VA
[3]National Institute of Aerospace, Hampton, VA
[4]Science Systems and Applications, Inc., Hampton, VA
[5]Scripps Institution of Oceanography, University of California San Diego, La Jolla, CA
[6]Pacific Marine Environmental Laboratory, NOAA, Seattle, WA, USA
[7]Cooperative Institute for Climate, Ocean and Ecosystem Studies, University of Washington, Seattle, WA, USA
[8]Plymouth Marine Laboratory, Prospect Place, Plymouth, United Kingdom
[9]Department of Earth System Science, University of California, Irvine, CA, USA
[10]Applied Physics Laboratory, Air-Sea Interaction and Remote Sensing Department, University of Washington, Seattle, WA, USA
[11]Oregon State University, Corvallis, OR

*Correspondence to*: Kevin J. Sanchez (kevin.j.sanchez@nasa.gov) and Richard H. Moore (richard.h.moore@nasa.gov)

**Abstract.** Atmospheric marine particle concentrations impact cloud properties, which strongly impact the
amount of solar radiation reflected back into space or absorbed by the ocean surface. While satellites can provide a snapshot of current conditions at the overpass time, models are necessary to simulate temporal variations in both particle and cloud properties. However, poor model accuracy limits the reliability with which these tools can be used to predict future climate. Here, we leverage the comprehensive ocean ecosystem and atmospheric aerosol-cloud data set obtained during the third deployment of the North
Atlantic Aerosols and Marine Ecosystems Study (NAAMES3). Airborne and ship-based measurements were collected in and around a cold-air outbreak during a three-day intensive operations period from September 17-19, 2017. Cold-air outbreaks are of keen interest for model validation because they are challenging to accurately simulate, which is due, in part, to the numerous feedbacks and sub-grid scale processes that influence aerosol and cloud evolution. The NAAMES observations are particularly valuable
because the flight plans were tailored to lie along Lagrangian trajectories, making it possible to spatiotemporally connect upwind and downwind measurements with the state-of-the-art FLEXible PARTicle (FLEXPART) Lagrangian particle dispersion model and then calculate a rate of change in particle properties. Initial aerosol conditions spanning an east-west, closed-cell cloudy to clear air transition region of the cold-air outbreak indicate similar particle concentrations and properties. However, despite the
similarities in the aerosol fields, the cloud properties downwind of each region evolved quite differently. One trajectory carried particles through a cold-air outbreak, resulting in a decrease in accumulation mode particle concentration (-42%) and cloud droplet concentrations, while the other remained outside of the



cold-air outbreak and experienced an increase in accumulation mode particle concentrations (+62%). The variable meteorological conditions between these two adjacent trajectories result from differences in the local sea surface temperature altering stability of the marine atmospheric boundary layer because of the location of the Labrador Current. Further comparisons of historical satellite observations indicate that the observed pattern occurs annually in the region, making it an ideal location for future airborne Lagrangian studies tracking the evolution of aerosols and clouds over time under cold air outbreak conditions.

## 1 Introduction

Understanding marine aerosol, meteorological processes, and their impact on cloud properties is crucial for calculating the global radiative balance. The oceans cover approximately 70% of the earth and absorb 90% of incoming solar radiation reaching the ocean surface under most conditions (Li et al., 2006). Overlying clouds obstruct this heating by reflecting sunlight back into space. Both aerosol abundance and meteorology govern these processes and determine cloud lifetime and albedo. Aerosol can act as condensation nuclei for water vapor to condense and form cloud droplets, the number of which impact the cloud properties and radiative forcing (Comstock et al., 2004; Pawlowska and Brenguier, 2003; Platnick and Twomey, 1994; Sandu et al., 2008; Turner et al., 2007; vanZanten and Stevens, 2005; Warren et al., 1988). Despite the importance of marine clouds, there are few measurements of aerosol and clouds over the ocean due to the high cost of deploying appropriate platforms in the remote ocean. To fill this major void in measurements, model simulations are useful in understanding processes that occur in these remote areas; however, the simulated values still require measurements for model validation (Ackerman et al., 2000; Golaz et al., 2011; Grabowski, 2001; Khairoutdinov and Randall, 2001; Nissanka et al., 2018; Seinfeld et al., 2016; Suzuki et al., 2013; VanZanten et al., 2011; Wyant et al., 2015). Lagrangian measurements are particularly useful for validation, as they provide not only a measurement in time and space, but also additional measurements later in time that indicate how the initial measured properties have changed due to atmospheric processes. With this information, simulated values and their total derivatives can both be validated. Here we examine cloud and particle measurements along particle trajectories in and around a cold-air outbreak over the North Atlantic to identify impacts on particle evolution.

Cold-air outbreaks commonly occur in the North Atlantic and are characterized by an expansive area of closed-cell stratocumulus clouds that transition to form open-cell convective clouds. Cold-air outbreaks are of recent scientific interest because they provide well-posed cases to study regime-dependent cloud radiative properties and to improve model representation of the cloud evolution across these regimes (Field et al., 2014; Fletcher et al., 2016; Tselioudis et al., 2013). Such transitions have been extensively studied with modeling and observations in the subtropics, where the break-up of the closed-cell region is reportedly driven by the decoupling of the cloud layer from the surface (i.e., when an inversion separates the boundary layer into two distinct layers, impeding mixing between the layers) (Abel et al., 2017; Albrecht et al., 2016,



1995; Berner et al., 2013; Bretherton and Wyant, 1997; Christensen et al., 2020; Ghate et al., 2015; Lloyd et al., 2018; de Roode et al., 2016; Sandu and Stevens, 2011; Wood et al., 2011, 2017; Yamaguchi et al., 2017). There are two main processes that lead to the decoupling and stratocumulus-to-cumulus transition.

First, advection of cold Arctic air over warmer sea surface temperatures (SST) increases sensible and latent heating, causing the marine boundary layer depth to increase. With increased boundary layer depths and surface heating, the top-down circulation of marine stratocumulus clouds (driven by cloud top radiative cooling and cloud top evaporative cooling), can no longer extend to the ocean surface, initiating the decoupling of the marine boundary layer (Albrecht et al., 1995; de Roode et al., 2016; Wang and Feingold,

2009). Second, cloud-top entrainment of free tropospheric air warms and dries the decoupled cloudy layer as the below-cloud surface-coupled layer continues to moisten from the latent heating, thereby strengthening the inversion between the two layers (Albrecht et al., 1995; Bates et al., 1998a; Bretherton and Wyant, 1997; Zhou et al., 2015). The progression of these two processes decreases the altitude of the lifting condensation level of rising air and increases the altitude of the lifting condensation level of sinking

air, respectively, causing the stratocumulus cloud base to rise until it has completely evaporated, while also forming convectively driven cumulus clouds with lower cloud bases. The progression rate of the stratocumulus-to-cumulus transition is heavily dependent on the strength of the decoupling in the MBL and how fast the decoupled cloud layer dries through cloud-top entrainment and precipitation. Both cloud-top entrainment rates and precipitation rates are heavily influenced by the cloud layer aerosol properties (Abel

et al., 2017; Ackerman et al., 2004; Albrecht, 1989; Berner et al., 2013; Hill et al., 2009; Jiang et al., 2006; Lu and Seinfeld, 2005; Stevens et al., 2005; Twomey, 1977; Yamaguchi et al., 2017).

The sensitivity to cloud-top entrainment and precipitation is what makes simulating the stratocumulus-to-cumulus transition in cold-air outbreaks difficult for models (Abel et al., 2017; Wang and Feingold, 2009; Xiao et al., 2012). The influence of aerosol on precipitation is well explained by the second-aerosol indirect

effect. In short, comparing two clouds with the same amount of cloud liquid water, one being fed with lower cloud condensation nuclei (CCN) number concentrations results in fewer and larger droplets relative to another cloud fed with greater CCN number where the outcome is more numerous and smaller droplets (Twomey, 1977). The larger droplet sizes can precipitate more easily, removing water from the cloud and decreasing its lifetime. The influence of the cloud layer aerosol and CCN concentrations on the drying from

entrainment are less intuitive. For example, large-eddy simulations of CCN-poor stratocumulus clouds have shown that sedimentation of large droplets near cloud-top can reduce the amount of cloud-top evaporation and overall drying of the decoupled cloud layer, prolonging the life of the cloud (Ackerman et al., 2004; Bretherton et al., 2007; Hill et al., 2009). Similarly, without sedimentation, cloud-top evaporation could lead to the break-up of stratocumulus clouds by gradually drying out the cloud layer (Bretherton and Wyant,

1997). Entrainment, however, alters the CCN concentration in the cloud layer through dilution since the



free troposphere CCN concentration is typically less than the marine boundary layer concentration. Any CCN lost through dilution are not replenished because the cloud layer is decoupled from the ocean surface source. As the CCN concentration decreases, fewer droplets are formed in subsequent ascents in the stratocumulus cloud, leading to larger droplets and drizzle formation. In addition, deepening of the stratocumulus layer could also enhance drizzle formation due to the increase in the liquid water content. The drizzle causes a positive feedback effect, as it leads to further removal of CCN through collision-coalescence (Chen et al., 2011; Lu and Seinfeld, 2005; Wood, 2007). This feedback ultimately decreases the cloud layer drying rate from entrainment and enhances the drying rate from precipitation. In the open-cell region after the transition, the marine boundary layer CCN concentration is relatively low due to the precipitation scavenging and collision-coalescence (Berner et al., 2013; Sharon et al., 2006; Terai et al., 2014; Wang and Feingold, 2009; Wood et al., 2011, 2017) that occurred upwind in the closed-cell region. Consequently, these processes cause relatively low cloud droplet concentrations, low cloud albedo, and increased precipitation in the clouds that form in the closed-cell region.

Marine particle concentrations can also vary significantly depending on their source. When present, particles from continental and pollution sources typically account for a significant majority of the existing particle concentration in marine regions (Coggon et al., 2012; Yang et al., 2016). In a recent study, Saliba et al. (2020) indicates that there are almost always some continental and pollution particles in the North Atlantic marine boundary layer, accounting for >10% of the total organic and sulfate mass (up to 50% and 80%, respectively) under what is otherwise identified as clean conditions. The origin of these particles is likely through entrainment into the marine boundary layer from the free troposphere (Shank et al., 2012). In the absence of direct advection of continental and pollution particles, the main sources to the marine boundary layer are sea spray from bubble bursting and wave breaking, as well as biogenic sources (Bates et al., 1998a; Covert et al., 1992; Frossard et al., 2014; De Leeuw et al., 2011; Modini et al., 2015; Murphy et al., 1998; Quinn et al., 2000, 2014; Rinaldi et al., 2010; Sievering et al., 1999; Thorpe, 1992; Warren and Seinfeld, 1985). Marine phytoplankton are an important source of volatile organic compounds (VOCs) that are released into the marine boundary layer and then oxidized to form low-volatility compounds that partition to the aerosol phase. This new aerosol mass can form onto existing particles or, in the absence of a meaningful amount of pre-existing aerosol surface area, nucleate new particles (Andreae and Crutzen, 1997; Ayers et al., 1997; Bates et al., 1998b; Chen and Jang, 2012; Clarke et al., 2013; De Reus et al., 2000; Sanchez et al., 2018; Veres et al., 2020). Despite precipitation acting as a major sink for marine particle concentrations, the removal of aerosol can result in optimal conditions for new particle formation and high concentrations of small Aitken mode particles (Clarke, 1993; Clarke et al., 1999; Raes et al., 1997; Russell et al., 1998; Sanchez et al., 2018; Seinfeld and Pandis, 2006; Thornton et al., 1997; Yue and Deepak, 1982).





Such feedbacks between meteorology and aerosol sources and sinks can cause challenges in determining

when and where marine particles will form.

Marine emissions of VOCs from the North Atlantic Ocean vary seasonally with the phytoplankton bloom cycle (Bell et al., 2021). The yearly cycle begins with the ocean mixed layer deepening in the winter, mixing nutrients up to the ocean surface and diluting the phytoplankton predators, which initiates the start of the phytoplankton bloom (Behrenfeld and Boss, 2018). The phytoplankton biomass continues to accumulate

throughout the spring, where the increasing sunlight continues to drive photosynthetic primary productivity, leading to the peak of the phytoplankton bloom in the late spring (Balaguru et al., 2018; Boss and Behrenfeld, 2010). The measurements presented in this study are collected in mid-September, after the peak in the phytoplankton bloom, but before the bloom significantly dissipates later in the fall and winter through the decrease of photosynthetically available radiation (Behrenfeld and Boss, 2018).

In this study, with careful planning and a bit of luck, we have identified two adjacent air masses with differing meteorological conditions: one initially in the closed-cell region of a cold-air outbreak and the other initially in a cloud-free regime. Comprehensive NAAMES ship and aircraft measurements of aerosol and clouds were made downwind of both cases. While the initial measurements are in different meteorological regimes, they are close in proximity, have similar upwind sources, and similar initial aerosol

properties. Our analysis focuses on identifying what processes led to the observed downwind differences in aerosol properties. In addition, ocean and atmospheric physical properties are examined to identify why the local meteorology varied significantly. Furthermore, we used historical satellite data to show how consistent the observed ocean and atmosphere conditions are in the region that led to the diverse meteorological forcing and influence on aerosol properties in close proximity. These measurements from

NAAMES and historical satellite analysis are informative to those seeking to perform similar parallel Lagrangian studies in future campaigns.

## 2    Methods

Here, we describe the measurements made on-board the *R/V Atlantis* ship and C130 aircraft used for the NAAMES3 case studies as well as the satellite and model reanalysis data products that are used to explain

the observations and provide context for future studies. The NAAMES3 campaign was conducted during the transitional decline in phytoplankton biomass (September 2017). A detailed description of all NAAMES campaigns can be found in Behrenfeld et al. (2019).

### 2.1  C130 Airborne Measurements

On the C130, aerosols were sampled through a low turbulence, isokinetic inlet and passed to each

instrument. Total particle concentration is measured with two Condensational Particle Counters at 1 Hz



with lower cut-off sizes of 10 nm (CPC Model 3772, TSI Inc., St. Paul, MN) and 3 nm (CPC Model 3025, TSI Inc., St. Paul, MN). A Laser Aerosol Spectrometer (LAS Model 3340, TSI Inc., St. Paul, MN) measures particle optical diameter distributions between 100 nm and 3500 nm at 1 Hz. The LAS particle concentration is integrated to identify the number of particles greater than 100 nm. Submicron particles are

also analyzed with a high-resolution time-of-flight aerosol mass spectrometer (AMS, Aerodyne Research Inc., Billerica, MA) (DeCarlo et al., 2006) that measures non-refractory inorganic (sulfate, ammonium, nitrate, chloride) and organic components at 30 second intervals. To identify anthropogenic pollution, refractory black carbon particle mass is measured with a Single Particle Soot Photometer at 0.1 Hz (SP2, DMT, Boulder, CO) and carbon monoxide mixing ratio is measured with a $CO/CO_2$ gas analyzer that

employs a cavity enhanced absorption technique at 1 Hz (LGR, San Jose, CA). CPC, LAS, AMS and SP2 measurements are all reported with respect to standard temperature and pressure (273.15 K, 1013 hPa). A Cloud Droplet Probe (CDP, DMT, Boulder, CO) measured cloud droplet size distributions for droplets ranging from 2 to 50 µm in diameter and a Cloud Imaging Probe (CIP, DMT, Boulder, CO) measured cloud droplet size distributions for droplets ranging from 50 to 1600 µm in diameter. The nadir-pointing High-

Spectral Resolution Lidar (HSRL) measured the vertical curtain of aerosol backscatter coefficient (532 nm wavelength) as well cloud top heights (Hair et al., 2008). The dimethylsulfide (DMS) mixing ratio was measured using a Proton Transfer Reaction Time of Flight Mass Spectrometer (PTR-ToF-MS) at 1 Hz (Müller et al., 2014). Air temperature was measured with a non-deiced Total Temperature Sensor (Model 102, Rosemount, St. Louis, MO), and total and static pressure was measured with a flush-mounted static

pressure sensor and total pressure sensor (MADT 2014, Rosemount, St. Louis, MO). Wind components were measured with differential pressure measurements across a 5-hole pressure port system configured on the C130 nose radome. Wind-component measurements were corrected for aircraft altitude and inertial motion.

### 2.2 *R/V Atlantis* **Measurements during NAAMES**

*R/V Atlantis* aerosol instruments sampled air through a temperature-controlled, isokinetic inlet mounted on the forward O2 deck of the ship (~18 m above sea level). Collected air was subsequently dried using silica gel diffusion dryers before the flow was distributed to the instruments. A 1.0 µm sharp cut cyclone (SCC 2.229, BGI Inc. US) removes large, coarse mode particles (mainly sea-salt) in order to capture only the submicron aerosol fraction. Particle number concentrations above ~13 nm and ~5 nm were measured using

two Condensation Particle Counters (CPC 3010/3785, TSI Inc., St. Paul, MN), while a Scanning Electrical Mobility Sizer (SEMS, Model 138, 2002, BMI, Hayward, CA) measured dry particle size distributions (0.01-0.9 µm diameter). A Differential Mobility Particle Sizer (DMPS, University of Vienna) (Winklmayr et al., 1991) was used to measure the number size distribution of dry submicron (0.02–0.8 µm diameter)



ambient particles when SEMS measurements were unavailable. Accumulation mode particle number

concentrations (diameters > 100 nm) are calculated by integrating the portion of the SEMS and the DMPS

size distributions above 100 nm. A Single Particle Soot Photometer (SP2, DMT, Boulder, CO) is used to

measure refractory black carbon mass concentration. A second AMS (same model as the one on the C130)

is on the *R/V Atlantis* to measure non-refractory inorganic (sulfate, ammonium, nitrate, chloride) and

organic particles (DeCarlo et al., 2006). A dual-flow-loop two-filter radon ($^{222}$Rn) detector provides

information on air mass origin, and specifically, continental influences (Whittlestone and Zahorowski,

1998). DMS is measured with a chemical ionization mass spectrometer (Bell et al., 2013, 2015).

### 2.3 **FLEXPART Back Trajectories**

We use the FLEXible PARTicle dispersion model (FLEXPART; (Stohl et al., 2005)) to assess air mass

back trajectories and to identify cases in which we can link upwind particle measurements to subsequent

downwind measurements (Zhang et al., 2014). Ten-day back trajectories with 6-hour intervals are computed

for the *R/V Atlantis* cruise track every hour and the C130 flight path every 20 minutes. The Global Forecast

System (GFS) and its final analysis (NCEP/NWS/NOAA/USDC, 2000) with 3-hour resolution, 1° 

horizontal resolution, and 26 vertical levels are used to drive all simulations. Each simulation consists of

ten thousand passive particle tracers released at the *R/V Atlantis* or C130 location, and the advection and

dispersion of the particles are simulated backwards in time. Positions of these tracers are used to compute

a gridded distribution of particle residence times (i.e., the average time an air parcel stays within a model

grid cell). In our analysis we average the latitude and longitude of the 6-hour interval gridded distributions

to produce a trajectory line. The vertical structure of the residence time is column-integrated over only the

vertical levels that are completely or partially within the MBL based on GDAS MBL heights. Remaining

vertical levels were excluded from analysis. More details about the FLEXPART trajectories and the

application to study the impact of marine biogenic particle on cloud can be found in Sanchez et al. (Sanchez

et al., 2021).

### 2.4 **Satellite and Model Data Products**

Merged satellite products, at 0.25° horizontal resolution (chlorophyll-a and cloud fraction), are obtained

from the GlobColour project (Maritorena et al., 2010; Maritorena and Siegel, 2005). Here, we use

chlorophyll-a (CHL-a) as a simple proxy for phytoplankton biomass (Behrenfeld et al., 2016; Lyngsgaard

et al., 2017; Meskhidze and Nenes, 2010; Pastor et al., 2013). GOES-13 visible satellite imagery is used to

provide a perspective of cloud conditions relevant for the C130 flights, the *R/V Atlantis* ship track and the

FLEXPART back trajectories. Sea surface temperature from Level 4 data products, derived from optimally





interpolated multi-sensor high-resolution datasets, are available from the Group for High Resolution Sea
Surface Temperature (GHRSST) (ABOM, 2008). Finally, 6-hour instantaneous horizontal wind vectors at
985 mb are obtained from the Modern-Era Retrospective analysis for Research and Applications, version 2
(MERRA-2) and used to determine seasonal patterns in wind direction (Gelaro et al., 2017; Global
Modeling And Assimilation Office, 2015). Satellite derived surface currents are obtained from Ocean
250    Surface Current Analysis Real-time (OSCAR) (Bonjean and Lagerloef, 2002; ESR, 2007).

## 3   Results

This study focuses on aerosol and cloud measurements in and around a cold-air outbreak event that occurred
over the western North Atlantic Ocean on 17 - 19 September 2017 (depicted by the GOES visible imagery
shown in Figure 1). The NAAMES3 campaign ship cruise track is given by the thick, grey line, and
255    instantaneous ship (yellow points) and aircraft positions (red lines when altitude < 3 km and cyan lines
when altitude > 3 km), coincident with each satellite image (± 1 hour), in Figure 1. The large-scale wind
direction of the cold-air outbreak tended to be northwesterly, which is consistent with the observed
southeastward transition from closed to open-cell clouds and advection of Arctic air down the Labrador Sea
to the North Atlantic region. Aerosol measurements were collected on the *R/V Atlantis* as it transected the
260    open-cell region of the cold-air outbreak (Figure 1d, e, S1) and then passed under an adjacent mostly clear-
sky region to the southwest (Figure 1f-j, S1). Two complementary C130 flights were conducted on 17 and
19 September (from here on referred to as FLT17 and FLT19), where the FLT17 flight track was likely
upwind of both the *R/V Atlantis* and the FLT19 flight track, as shown by FLEXPART trajectories. During
FLT17, the *R/V Atlantis* was positioned at a single measurement station for conducting over the side
operations before voyaging southwest at 18:15 UTC 17 September 2017. During FLT19, the ship was
underway with a southwest heading.

### 3.1 Summary of *R/V Atlantis* and C-130 Observations

The time series of the *R/V Atlantis* particle and DMS concentrations, and particle composition from 17
through 19 September 2017 is shown in Figure 2, where the two highlighted boxes emphasize the transitions
in measured aerosol regimes. The first transition is due to the passing of an occluded front with a region of
open-cell clouds following behind, as seen by GOES images in Figure 1a-e. The second transition involves
the ship entering a region influenced by continental air, as evident from the elevated radon concentrations
(> 500 mBq m$^{-3}$) (Figure 2d). Between these two transitions the ship remains in the system of open-cell
clouds (Figure 1a-e, supplementary Figure S1). In this open-cell region, the particle mass concentration
increases as the *R/V Atlantis* travels southwest.


The aircraft measurements from FLT17 are conducted during 1025-2035 UTC on 17 September, upwind of the ship and FLT19 measurements conducted hours to days later, as discussed in detail in Sections 3.3. The black carbon concentration on both flights is consistent with previously identified clean marine conditions (<25 ng m$^{-3}$) (Saliba et al., 2020; Sanchez et al., 2021) and similarly, the observed carbon

monoxide is consistent with observations made at an eastern North Atlantic research facility that is dominated by clean marine conditions (<130 ppb) (Zheng et al., 2018) for near surface measurements made on both flights (Figure S2 and S3). Carbon monoxide has a lifetime of about 1 month (Seinfeld and Pandis, 2006) and therefore, is normally elevated in the North Atlantic relative to polar regions and other remote ocean regions that are not as often directly upwind of continental regions.

### 3.1.1   17 September 2017 Flight (FLT17)

Figure 3 shows C130 marine boundary layer particle, cloud droplet number concentrations (CDNC), and HSRL backscatter coefficient as a function of longitude for the roughly east-west flight paths shown in Figure 1c-e. After arriving at the ship, the aircraft initially transited west at low altitude making in-situ measurements before spiraling up to high altitude near 57°W and retracing the same flight path with the

HSRL. Because the aircraft was continually ascending and descending throughout the marine boundary layer during the in-situ sampling flight portion, we group the data into statistical boxes that represent the near-surface horizontal clear air legs (~0.1 km altitude) for the aerosol shown in Figure 3b, c. Similarly, the cloud measurements (Figure 3d-f) are grouped for each in-cloud horizontal leg (~1.0-1.5 km altitude) and also the vertical profile at ~39.5°W.  The nature of collecting CDNC along a linear flight path can introduce

a bias if measurements are disproportionately collected on cloud edges (downdraft regions) or cloud cores (updraft regions). For this reason, Table 1 includes several summary CDNC and updraft velocity statistics for each cloud leg. The CDNC geometric mean limits the influence of outliers relative to a simple arithmetic mean, while the updraft weighted CDNC removes the inclusion of measurements in downdrafts where CDNC may be decreasing due to evaporation and finally, the 90$^{th}$ percentile likely represents an

approximate value of the CDNC in an adiabatic updraft core. The weighted updraft is also included to show that the change in CDNC is not simply due to a change in updraft velocity. For example, the weighted updraft velocity at 53.0°W and 44.7°W are 0.34 and 0.33 m s$^{-1}$, respectively, but the CDNC between the two locations vary by a factor of 3-5. Figure 3a shows the C130 altitude from the low-altitude aircraft flight track overlaid on the high-altitude remote HSRL backscatter curtain over the same flight track, which was

obtained during a high-altitude, remote sensing leg that was conducted immediately after the near-surface in-situ measurement flight legs with minimal delay. It is apparent from Figure 3e that the western-most two in-cloud legs (at 53°W and 55°W) tended to have average CDNCs that were approximately 2-4 times higher than those observed on the eastern part of the track. The CIP measurements in Figure 3d show that the





clouds were precipitating, with peak precipitation sized droplets near the open to closed-cell transition,
indicating there was removal of aerosol and cloud droplets through precipitation scavenging. This is
consistent with previously published observations (Abel et al., 2017; Mechoso et al., 2014; Wood et al.,
2011) and simulations (Mechoso et al., 2014; Wang and Feingold, 2009; Yamaguchi et al., 2017), showing
reductions in CDNC ranging from 50% to 90%. These two regimes are identifiable in both the GOES
imagery (Figure 1), as well as the HSRL curtain (Figure 3a), where the highest observed CDNCs are in the
closed-cell cloud regime to the west, while the lowest CDNCs are in the open-cell cloud regime to the east.
Intermediate CDNCs are apparent in the transitional region in between. While the overall aerosol number
concentration ($N_{>3nm}$) does not have a clear trend throughout this period, the accumulation mode aerosol
number concentration ($N_{>100nm}$) (Figure 3c) and bulk organic and sulfate particle mass concentration (Figure
3b) are lower on the eastern part of the flight, consistent with the lower CDNCs. There is no clear trend in
the observed LWC (Figure 3f), other than the fact that the highest LWCs were higher in the open-cell
region, where cloud-tops were higher. The HSRL particle backscatter coefficient measurements show the
westernmost near-surface horizontal leg occurred in a cloud-free zone west of the closed-cell cloud regime
(Figure 3a). Notably, the observed near-surface particle concentrations and the back trajectories initialized
in the cloud-free and closed-cell regimes are similar (53°W-59°W in Figure 3b, c; supplementary Figure
S4), demonstrating that both regimes contain similar particle populations and have a similar source, despite
differences in meteorology. In the remaining sections, we explore the cause of the differing adjacent
meteorological conditions and their impact on downwind aerosol and cloud properties.

### 3.1.2    19 September 2017 Flight (FLT19)

Figure 4 shows C130 marine boundary layer aerosol, cloud droplet measurements, and HSRL back scatter
coefficient as a function of longitude (similar to the FLT17 case in Figure 3) for the flight paths shown in
Figure 1g-j. Similar to Table 1, Table 2 includes CDNC and updraft velocity statistics for each cloud leg,
showing the trend along the flight path. Note the portion of FLT19 shown in Figure 1i has a significantly
larger northward component compared to the low-altitude portion of the flight path shown in Figure 1j,
causing the higher density of vertical profiles and horizontal leg measurements between 37°W and 40°W
in Figure 4 and Table 2. Figure 1 satellite images and Figure 4a HSRL backscatter measurements show that
none of the near-surface or in-cloud measurements were in the closed-cell region of the cold-air outbreak.
Similar to the FLT17 case, the HSRL curtain covered roughly the same horizontal location as the C130
surface measurements in Figure 4 (data are not reported here for two periods where the aircraft deviated
from the subsequent low-altitude flight track). For FLT19, the high-altitude remote sensing leg was
conducted immediately before the in-situ surface measurements. A key feature to note in Figure 4 is the
significant change in average particle concentration and CDNC to the west of 40°W. Also, the clouds


associated with the high particle and CDNC are non-precipitating (Figure 4d). Further analysis in section 3.3.2 suggests that the higher particle and CDNC region is downwind of the cloud-free region measured on FLT17 and not the cold-air outbreak.

**3.2 C130 Vertical Profile Stability**

Figures 5a and 5b present vertical profiles of potential temperature corresponding to the aircraft inline ascents and descents shown in Figure 3a and Figure 4a for FLT17 and FLT19, respectively. Only continuous vertical profiles are included in Figure 5, though some are excluded to prevent substantial overlap of multiple similar vertical profiles. The square points at the sea surface (0 km altitude) represent
the satellite measured SST. The color of each point in Figure 5 represents the longitude at which the measurement was made. For FLT17, shown in Figure 5a, the marine boundary layer is neutrally buoyant to the west and becomes increasingly decoupled farther to the east (with the exception of the western most vertical profile at 58°W). This evolution in the profile stability and particle concentration is consistent with cold-air outbreaks where processes in the closed-cell region cause the marine boundary layer to decouple,
leading to the transition from closed to open-cell clouds (Abel et al., 2017; Albrecht et al., 1995; Christensen et al., 2020; de Roode et al., 2016; Sandu and Stevens, 2011; Wood et al., 2011; Yamaguchi et al., 2017). None of the vertical profiles from FLT19 are in the closed-cell region, but the profiles to the west, nearest the closed to open-cell transition of the cold air outbreak are decoupled and stable (Figure 5b), consistent with observations in FLT17. However, the neutrally buoyant boundary layers to the east of 40°W are not
consistent with this pattern. These boundary layer profiles are neutrally stable, despite being in an area of cumulus clouds because the eastern portion of the flight is not actually part of the open-cell region of cold-air outbreak or even downwind of it (discussed in section 3.3.2), and is therefore not influenced by meteorological processes associated with the cold-air outbreak.

The FLT17 stable vertical profile at about 58°W (Figure 5a) is from the clear sky region that is to the west
of the closed-cell region (Figure 1d). This vertical profile is very stable and strongly decoupled. The cause of the differing atmospheric stability in this region is the low SST (Figure 5a), which cools and stabilizes the marine boundary layer, preventing the vertical transport of water vapor and cloud formation. For almost all the other vertical profiles in Figure 5, the SST is greater than the atmospheric temperature near the sea surface, consistent with the processes of advection of cold-air over warm SST that lead to the stratocumulus-
to-cumulus transitions in cold air outbreaks (Albrecht et al., 1995; de Roode et al., 2016). The neutrally stable vertical profiles with higher particle concentrations and CDNC in FLT19 (Figure 5b and 4b, c, e) are actually downwind of this same clear sky region (further discussed in section 3.3.2). The reason for the characteristic differences in the measurements downwind of the clear-sky and cold-air outbreak regions,





despite their initial close proximity, is discussed in the following section examining relevant FLEXPART
back trajectories.

### 3.3 Particle Evolution Along FLEXPART Trajectories

### 3.3.1 FLT17 to *R/V Atlantis* Trajectories

The *R/V Atlantis* ship track was well placed downwind of the portion of FLT17 shown in Figure 1c-d such
that spatially averaged FLEXPART back trajectories initialized from the ship position overlap in time and
space (within 3 hours and 1° latitude and longitude) with the C130 near-surface measurements (Figure 6).
This overlap enabled the comparison of measured particle properties between the C130 and *R/V Atlantis* to
identify changes over time. In Figure 6, available (daylight only) visible GOES images of the 2°×2° area
are shown and aligned with the 6-hourly trajectory interval that overlaps the up-wind FLT17 measurements
or at the initialization location of the back trajectory. These images are shown to identify if measurements
were made in the cloud-free, open or closed-cell regions. Upwind, near-surface C130 measurements within
1° latitude and longitude and 3 hours of the back trajectory point are averaged and subtracted from the
downwind ship measurements (shown in Table 3) to represent changes in the particle properties during
transport. In general, both particle mass and number concentration decreased over the trajectory. However,
the difference in particle concentrations is less than the standard deviation for some cases and therefore
within the measured variability. This occurs mostly when the trajectory length is short and therefore
changes in particle properties over the short time period are minimal. Also, particle number concentrations
have somewhat smaller standard deviations than particle mass concentrations, due to the greater sampling
rate and smaller instrument error of the CPC and LAS, compared to the AMS. The time periods used for
the *R/V Atlantis* measurements in this comparison are from 10-minute averages at the back trajectory
initialization time (highlighted in Figure 2). The magnitude of the decrease in particle number concentration
generally increased for the longer trajectory times (8-16 hours) with the exception of the last trajectory
(shown in Figure 6e). This last trajectory is different in that its path is near the transition between the cloud-
free and closed-cell regions (Figure 1d, 6e). The 2°x2° satellite image at the *R/V Atlantis* location (the
trajectory initialization location) shows open-cell clouds, which could mean clouds had formed along the
trajectory or that the measurement is some combination of both air from the cloud-free region and the
closed-cell region upwind. Either way, the different meteorological processes in the initially cloud-free and
closed-cell region of the cold air outbreak likely resulted in the decreased the overall rate in aerosol particle
removal between the two measurements relative to upwind trajectories that were clearly in the cold-air
outbreak (Table 3).

### 3.3.2 FLT17 to FLT19 Trajectories



FLEXPART back trajectories are also initialized with FLT19 near-surface measurements, which are downwind of the FLT17 flight path. The initial easternmost aerosol measurements from FLT19 (Figure 1i, 4b-e) have the highest aerosol mass and number concentrations, and CDNC, for the low-level in-situ measurements shown in Figure 4. Examining the FLEXPART back trajectories initialized at this location, we find that the upwind source of the region with high particle concentrations transported from the cloud-free stable boundary layer observed in FLT17. Figure 7 highlights this shift in upwind source with trajectories that cross the cloud-free region (Figure 7a), the transition zone between the closed-cell and cloud-free regions (Figure 7b), and the closed-cell region of the cold-air outbreak (Figure 7c). Since the Figure 7b trajectory aligns the transition zone between the closed-cell and cloud-free region upwind and the transition zone between low and high particle concentrations downwind, this trajectory is used to compare both pairs of measurements on either side of these zones. Table 3 shows calculations of the difference in downwind (FLT19) and upwind (FLT17) observed particle loadings, which we use to infer differences in particle evolution from the cloud-free region leading up to the observed FLT19 high particle concentrations and the evolution of particles from the closed-cell region to the observed FLT19 low particle concentrations. Over approximately 48 hours, particles that advected from the closed-cell region of the cold-air outbreak in FLT17 and to the open-cell region between 37.8°W and 38.5°W in FLT19, show the average organic and sulfate concentrations significantly decreased (100% and 44%, respectively) in the cold-air outbreak (Table 3). The preferential removal of organic mass is unexpected as sulfate is more hygroscopic and therefore, more likely to activate to form cloud-droplets and be lost through precipitation processes. It is possible the sulfate mass resided in smaller particles that are less likely to form cloud droplets or that there are differences in the replenishment of mass via marine or entrained sources, similar to previous particle formation events observed in cloudy marine environments (Clarke et al., 1999; Hegg et al., 1990; Perry and Hobbs, 1994; Sanchez et al., 2018). Sanchez et al. (2021) showed that organic aerosol mass is strongly correlated with in-water biological activity, while sulfate aerosol mass is only weakly correlated. They attributed the difference to the relatively short lifetime of the organic aerosol precursors like isoprene and monoterpenes (minutes to hours) versus sulfate precursors like DMS (>1-2 days). We speculate that the various source strength and precursor residence time may play a role in the resulting shift in organic to sulfate mass ratio. Additionally, the $CN_{>10nm}$ and $CN_{>100nm}$ particle concentrations both decreased by 40-50% (Table 3). Conversely, over the same approximate 48-hour period, particles that advected from the cloud-free region in FLT17 to the cumulus cloud region east of 37.8°W in FLT19, show a decrease in organic particle concentration within the observed variability (10%) and an increase in sulfate particle concentration (37%) (Table 3). The $CN_{>10nm}$ and $CN_{>100nm}$ particle concentrations decreased by 42% and increased by 62%, respectively (Table 3). The in-cloud CIP measurements from FLT19 (Figure 4d, in the region downwind of the cloud –free region, east of 37.8°W) show essentially no precipitation size





droplets; however, we cannot rule out precipitation occurring along the trajectory between the FLT17 cloud-
free region and downwind FLT19 measurements. The differences in the meteorology along these
trajectories suggests that the trajectory outside the cold-air outbreak likely had less precipitation and
corresponding collision-coalescence scavenging, resulting in the observed differences in the particle
property evolution.

**3.4 Ocean SST Control of Atmospheric Stability**

Averaged satellite SST measurements for September 2017 (corresponding to the NAAMES3 mission) show
a local minimum in SST (highlighted by the black box in Figure 8a) in the cloud-free region to the west of
the observed closed-cell region in FLT17 (Figure 1c). The southerly flow of the Labrador Current, bringing
cold water from the Arctic Ocean (Figure S5) causes the observed minimum in SST. Figure 8b shows the
spatially collocated minimum in the average cloud fraction (CF), consistent with the visible satellite images
in Figure 1. Figure 8c shows the linear regression between the September 2017 averaged SST and CF from
the boxed areas in Figures 8a and 8b. Each point in Figure 8c represents a pixel-by-pixel comparison from
the boxed region of Figure 8a and 8b. The spatial correlation between the SST and CF suggests that the
SST strongly influences the CF in this region during NAAMES3, likely because the air at the surface cools
and stabilizes the atmospheric marine boundary layer (consistent with the stable vertical profile and SST at
58°W shown in Figure 5a). A stable marine boundary layer prevents vertical transport of water vapor, thus
trapping moisture near the surface, inhibiting cloud formation in the region, and producing the persistent
low CF relative to the surrounding area. Our Lagrangian analysis in section 3.3 demonstrates the significant
difference in meteorological impacts on cloud properties and particle concentrations downwind of this very
stable boundary layer and the adjacent closed-cell region of a cold-air outbreak.

To identify how consistently the SST influences the CF and meteorology in this region, CF and SST are
compared for each month using satellite observations from August 2008 to June 2019. The correlation
(Pearson's Coefficient) between the CF and SST is consistently highest in the late summer, peaking in
September, consistent with NAAMES3 analyzed case studies (Figure 8d). Even the wind direction from
MERRA-2, over the same ~11-year time period, is typically from the northwest (Figure S6), parallel to the
Labrador current, which is necessary for measurements in the North Atlantic to be downstream of the
highlighted area of interest in Figure 8. Based on these results and the Lagrangian cases shown, this region
is an ideal location to study the effect of cloud processing on marine boundary layer particle properties. The
proximity of the cloud-free and cold-air outbreak closed-cell region are close and therefore, likely often
containing inflows of air from similar sources, with similar particle properties. In addition, the difference
in meteorological conditions caused by the SST pattern in the region enables one to compare the evolution
of the particles initialized in these two different meteorological regimes over a Lagrangian trajectory. The



controlling SST forcing on cloud formation in this region of study suggests changes in North Atlantic cold-air outbreaks, due to changes in climate, could have major implications for cloud and particle properties in

the North Atlantic. In a previous modeling study Kolstad and Bracegirdle (2008) have shown cold air outbreaks in the North Atlantic are predicted to weaken in the late 21st century as the difference in the atmospheric temperature and SST is expected to decrease, and therefore will likely affect the influence cold-air outbreaks have on the evolution of cloud and aerosol properties.

## 4    Summary and Conclusions

Here, we examine the evolution of atmospheric aerosols and clouds along adjacent Lagrangian air mass trajectories over the North Atlantic during a cold air outbreak event in 2017. While particle number concentrations and mass composition are similar within the northwest (i.e., upwind) portion of the study region, the cloud conditions are very different with a notable clear air region above the Labrador Current and a stratiform cloud region immediately to the northeast over warmer waters. This clear-to-cloudy

transition is well captured by the NASA C-130 and HSRL lidar measurements from NAAMES, which confirmed the similar aerosol conditions across the transition region. Subsequent measurements downwind of these regions, which we connect back using FLEXPART model trajectories show that the aerosol fields evolve very differently depending on whether they started out in the clear or cloudy areas.

Along the trajectory just outside of the cold air outbreak, where the initial cloud-free air evolves to cumulus

clouds, we observe no changes in the organic mass, but an increases in sulfate mass (37%) and large particle (diameter > 100 nm) concentration (61%) over a 48 hour period (Table 3). We contrast this case with the adjacent trajectory that is inside the cold air outbreak, where the closed-cell stratiform clouds evolve and become open-cell cumulus clouds. For this case, we observe large reductions in aerosol particle and cloud droplet number concentrations as well as an approximate 100% depletion of the organic aerosol mass

concentrations over the 48 hour period (Table 3). Surprisingly, we observed only a 44% reduction in sulfate aerosol mass over this same trajectory. The change in the organic to sulfate mass ratio is unexpected as sulfate components are more hygroscopic and therefore, more likely to activate to form cloud-droplets and be removed through precipitation. It is possible that the sulfate resided in smaller particles or more of the sulfate was removed through precipitation and subsequently more sulfate particles were formed via gas

phase or in-cloud oxidation of $SO_2$. Sanchez et al. (2021) showed that organic aerosol mass is moderately correlated with in-water biological activity, while sulfate aerosol mass is only weakly correlated. They attributed the difference to the relatively short lifetime of the organic aerosol precursors like isoprene and monoterpenes (minutes to hours) versus sulfate precursors like DMS (>1-2 days). Given the localized area of the higher ocean chlorophyll and biological activity to the northwest (Figure S7), we speculate that while

both organic and sulfate aerosol species are rapidly depleted by wet scavenging along the cold-air outbreak trajectory (i.e., the closed-to-open-cell transition), the sustained secondary sulfate source is able to replenish



the sulfate aerosol concentration. Meanwhile, the organic aerosol does not recover over this period, which gives rise to the 100% decrease in organic mass observed during NAAMES.

This case study is ideal for future model simulation because the dramatic gradient in SST caused by the Labrador Current essentially bifurcates the atmosphere along two pathways for aerosol-cloud evolution. This results in different initial cloud conditions (clear vs. closed-cell clouds), but similar initial aerosol conditions. Meanwhile, the downwind conditions are similar for the clouds (cumulus clouds) but the aerosol number and speciated mass concentrations are quite different. We speculate that these differences are driven by different amounts of cloud processing along the adjacent trajectories. C130 measurements of marine boundary layer potential temperature profiles are consistent with cloud-top entrainment and increased sea surface sensible and latent heating leading to the stabilization of the marine boundary layer. In addition, increased precipitation and collision-coalescence decreases the particle and cloud droplet concentrations. Both of these processes, in turn, lead to the transition from stratocumulus to convective cumulus clouds (Abel et al., 2017; Albrecht et al., 2016; Christensen et al., 2020; de Roode et al., 2016; Sandu and Stevens, 2011; Wood et al., 2011, 2017; Yamaguchi et al., 2017). The adjacent cloud-free and closed-cell regions appear to be driven by the sharp gradient in SST caused by the Labrador Current. To better understand how prevalent these conditions set up in this region, we analyzed 11 years of satellite measurements and found persistently low cloud fraction over the Labrador Current next to an area of high cloud fraction that occurs annually around September. This shows that future studies could take advantage of this reoccurring pattern to obtain more statistics and track the Lagrangian evolution of atmospheric aerosols resulting from different initial meteorological conditions.

**Author Contribution**

Conceptualization, Methodology, and Writing - Original Draft: **KJS and RHM**; Software: **KJS and BZ**; Formal Analysis, Visualization: **KJS**; Supervision, Project administration, Funding acquisition: **RHM, MHB**; Data Curation: **GS, CC, SLL, PKQ, TSB, JP, TGB, ESS, MJB, LMR;** Writing - Review & Editing: **all authors**.

**Acknowledgments**

We thank the dedicated crew of the R/V Atlantis. Kevin J. Sanchez was funded by the NASA Postdoctoral Program. The authors also would like to acknowledge Raghu Betha, Derek Price, Derek Coffman, James Johnson, Armin Wisthaler, and Lucia Upchurch for collecting and reducing data. This work was funded by NASA grant NNX15AE66G, NNX15AF30G, NNX15AF31G and NSF grant NSFOCE-1537943. This is PMEL contribution number 5277. Bo Zhang and Hongyu Liu acknowledge the funding support from the NAAMES mission. The NAAMES dataset is archived in the NASA Atmospheric Science Data Center (ASDC; https://doi.org/10.5067/Suborbital/NAAMES/DATA001) and the SeaWiFS Bio-Optical Archive and Storage System (SeaBASS; https://doi.org/10.5067/SeaBASS/NAAMES/DATA001). Scripps measurements are available at https://library.ucsd.edu/dc/collection/bb34508432. Shipboard measurements are archived at https://seabass.gsfc.nasa.gov/. GlobColour data (http://globcolour.info) used in this study has been developed, validated, and distributed by ACRI-ST, France. GDAS data are available at



ftp://arlftp.arlhq.noaa.gov/pub/archives/gdas1/. The SST is acquired from the Global Australian Multi-Sensor Sea
Surface Temperature Analysis (ABOM, 2008). OSCAR ocean currents are available at https://podaac-
tools.jpl.nasa.gov/drive/files/allData/oscar/L4/oscar_1_deg (ESR, 2007). MERRA-2 data is available through
https://earthdata.nasa.gov (Global Modeling And Assimilation Office, 2015).

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



**Table 1. In-cloud leg measurements of CDNC (cm-3) and updraft velocity (w, m s-1) from the C130 flight on 17 September 2017 (Figure 3).**

| Longitude | CDNC | | | w |
| --- | --- | --- | --- | --- |
| | Geometric mean±std | Updraft weighted | 90th Percentile | Updraft weighted |
| 54.9°W | 67±2 | 69 | 90 | 0.51 |
| 53.0°W | 40±3 | 60 | 93 | 0.34 |
| 51.0°W | 23±2 | 20 | 62 | 0.38 |
| 48.9°W | 20±2 | 30 | 57 | 0.25 |
| 47.0°W | 16±3 | 15 | 67 | 0.37 |
| 44.7°W | 8±2 | 11 | 20 | 0.33 |
| 39.4°W | 11±2 | NA | 32 | NA |




**Table 2. In-cloud leg measurements of CDNC (cm-3) and updraft velocity (w, m s-1) from the C130 flight on 19 September 2017 (Figure 4).**

| Longitude | CDNC | | | w |
| | Geometric mean±std | Updraft weighted | 90th Percentile | Updraft weighted |
|---|---|---|---|---|
| 43.6°W | 33±2 | 43 | 51 | 0.68 |
| 42.3°W | 14±2 | 23 | 29 | 0.69 |
| 41.1°W | 11±2 | 24 | 31 | 0.70 |
| 39.9°W | 14±2 | 24 | 33 | 0.96 |
| 39.4°W | 14±2 | 27 | 33 | 0.90 |
| 38.7°W | 6±2 | 7 | 12 | 0.30 |
| 38.2°W | 31±2 | 50 | 64 | 0.74 |
| 37.7°W | 40±2 | 76 | 87 | 0.81 |




**Table 3. The absolute and relative difference between the downwind and upwind measurements for the cases in which 2-day FLEXPART back trajectories initialized by the R/V Atlantis or C130 location overlap with upwind C130 measurement locations (within 1° latitude and longitude and 3 hours), shown in Figures 6 and 7. The difference in absolute values of the particle composition and concentration are calculated as an average ± 1 standard deviation.**

| Figure Panel and upwind cloud conditions | Dt (hours) | Particle Composition (µg m$^{-3}$) | | | | Particle Concentration (cm$^{-3}$) | | | |
|---|---|---|---|---|---|---|---|---|---|
| | | $\Delta Org$ | $\dfrac{\Delta Org}{Org}$ | $\Delta SO_4$ | $\dfrac{\Delta SO_4}{SO_4}$ | $\Delta N_{>10nm}$[4] | $\dfrac{\Delta N_{>10nm}}{N_{>10nm}}$[4] | $\Delta CN_{>100nm}$ | $\dfrac{\Delta N_{>100nm}}{N_{>100nm}}$ |
| Figure 6 | | **Trajectories initialized at the _R/V Atlantis_** | | | | | | | |
| a$_{open-cell}$ | 8 | 0.04±0.16 | 0.50 | 0.01±0.09 | -0.08 | -47±154 | -0.24 | -7±51 | -0.26 |
| b$_{open-cell}$ | 11 | 0.00±0.18 | -0.01 | -0.03±0.13 | -0.24 | -107±203 | -0.35 | -6±68 | -0.15 |
| c$_{open-cell}$ | 14 | -0.07±0.19 | -0.41 | -0.06±0.08 | -0.43 | -213±293 | -0.57 | -23±21 | -0.47 |
| d$_{open-cell}$ | 16 | -0.02±0.21 | -0.14 | -0.08±0.10 | -0.46 | -295±225 | -0.65 | -45±18 | -0.72 |
| e$_{mixture}$[1] | 30 | -0.09±0.21 | -0.44 | -0.10±0.08 | -0.58 | -239±174 | -0.68 | -32±9 | -0.64 |
| Figure 7 | | **Trajectories Initialized at the C130** | | | | | | | |
| b$_{closed-cell}$[2] | 48 | -0.30±0.21 | -1.00 | -0.09±0.04 | -0.44 | -188±68 | -0.49 | -27±13 | -0.43 |
| b$_{cloud-free}$[3] | 48 | 0.02±0.29 | -0.10 | 0.08±0.05 | 0.37 | -223±53 | -0.42 | 32±17 | 0.61 |

[1]Upwind measurements are near the transition between cloud-free and closed-cell regions and therefore, downwind measurements possibly represent a mixture of the two regimes.

[2]Difference between the measurements in the FLT17 closed-cell region and FLT19 low particle concentration region. Specifically, FLT17 measurements in the closed-cell region at 53.5°W (Figure3, 7b) are compared to downwind FLT19 C130 surface measurements between 37.8°W – 38.5°W (Figure 4, 7b).

[3]Difference between the measurements in the FLT17 cloud-free region and FLT19 high particle concentration region. Specifically, FLT17 measurements in the cloud-free region at 57.8°W (Figure3, 7b) are compared to downwind FLT19 C130 surface measurements east of 37.8°W (Figure 4, 7b).

[4]_R/V Atlantis_ N$_{>13nm}$ measurements are used for comparison to C130 N$_{>10nm}$ measurements.








**Figure 1. GOES-East visible satellite imagery at two-hour intervals for 17 September 2017 (left column) and 19 September 2017 (right column). Purple lines outline the coast of eastern Canada and the southern tip of Greenland. Cyan and red lines represent the flight track at altitudes > 3km and < 3km, respectively, ±1 hour**

**from the satellite image time (shown on the y-axis). The yellow line represents the ship position ±1 hour from the satellite image time. The gray line is the entire ship track for the NAAMES3 campaign.**



**Figure 2.** The R/V Atlantis time series of (a) AMS non-refractory particle mass composition, (b) particle concentrations as a function of size, (c) DMS concentration, and (d) SP2 black carbon and radon concentration (d) over the time frame shown in Figure 1. Particle concentrations > 100 nm are measured with an SMPS with the exception of a ~1 day period (18 September) in which the SMPS was not operational. For this period, the DMPS measurements are shown. Black carbon measurements are not available for a ~12 hour period on 18 September when the SP2 was not operational. The two opaque boxed areas represent approximate periods in which airmass transitions occurred. The first transition was a result of the passing of an occluded front with a




region of open-cell clouds following (Figure 1a-e) and the second transition is a result of the ship entering a polluted air region. The red vertical lines mark the 10-minute period of measurements used in Table 3 comparisons.



**Figure 3: C130 particle measurements collected on the 17 September 2017 flight. The (a) C130 HSRL backscatter over laid with C130 altitude (magenta), (b) non-refractory particle mass composition, and (c) particle concentrations (N) for particles > 3 nm, > 10 nm and > 100 nm in diameter. In-cloud (d) precipitation from the CIP (diameter > 50 µm), and (e) CDNC (diameter range of 2-50 µm) and (f) LWC from the CDP are shown with box plots for each in-cloud leg. All measurements are presented as a function of longitude. The high-altitude HSRL back scattering measurements were made immediately after the surface measurements along the same horizontal flight track, but in reverse (Figure 1c-e). Note, while negative particle mass measurements are not possible, they are included to prevent exclusion of the negative bias in the AMS measurements.**

**Figure 4: C130 particle measurements collected on the 19 September 2017 flight. The (a) C130 HSRL backscatter over laid with C130 altitude (magenta), (b) non-refractory particle mass composition, and (c) particle concentrations (N) for particles > 3 nm, > 10 nm and > 100 nm in diameter. In-cloud (d) precipitation from the CIP (diameter > 50 µm), and (e) CDNC (diameter range of 2-50 µm) and (f) LWC from the CDP are shown with box plots for each in-cloud leg. All measurements are presented as a function of longitude. The**





high-altitude HSRL back scattering measurements were made immediately before the surface measurements
along almost the same horizontal flight track, but in reverse (Figure 1g-j). Any detours made at high latitude
from the horizontal surface flight path were excluded from the HSRL swath shown. Note, while negative
particle mass measurements are not possible, they are included to prevent exclusion of the negative bias in the
AMS measurements.



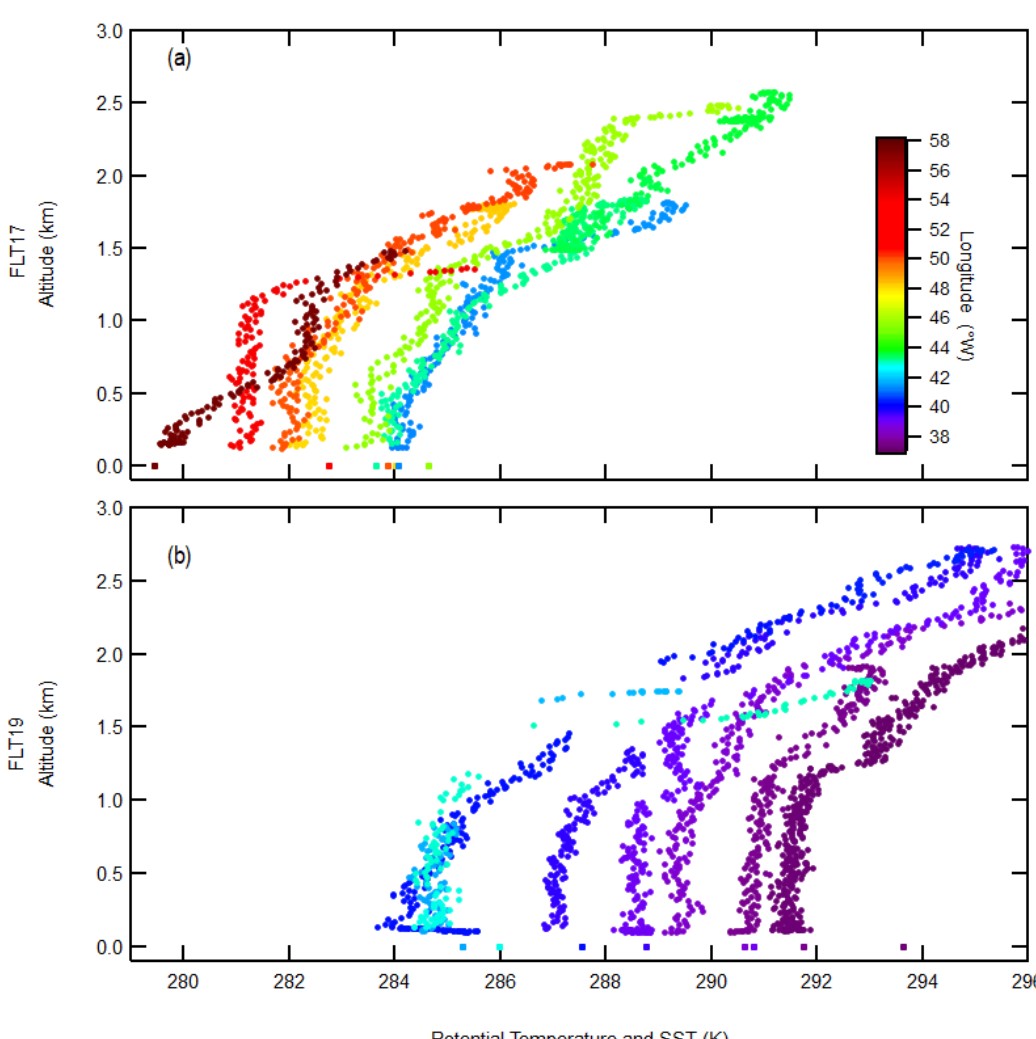

**Figure 5. Vertical profiles of the potential temperature from C130 flights that occurred on (a) 17 September 2019 and (b) 19 September 2019 (Figure 3a and 4a). Vertical profiles with horizontal in-cloud legs are excluded as well as a few others that have significant overlap with the presented vertical profiles. Surface measurements (at 0 km) represent the SST just below the lowest point in the vertical profile. The point color represents the longitude at which the measurement was made. Air mass transitions from opened-cell to closed cell clouds and from closed-cell to cloud-free air on the 17 September 2019 flight occurred at approximately 52 °W and 56 °W, respectively. None of the vertical profiles are in the closed-cell region for the 19 September 2019 flight.**






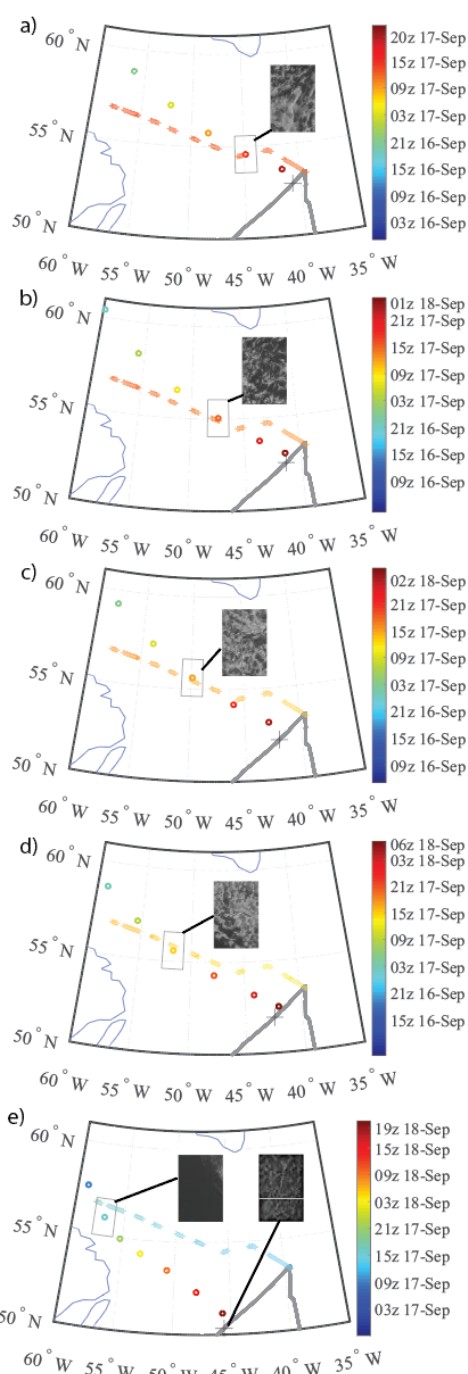

**Figure 6.** The location of FLEXPART back trajectories averaged over 6-hour intervals (shown as open circles) and low level (~0.1 km) C130 surface measurements (shown colored lines) are both colored as a function of time. The colorbar range is different for each panel and is defined by the initialization time and end time of a





**2-day FLEXPART back trajectory initiated at the R/V Atlantis. The initialization times for the back trajectories are (a) 23 UTC 17 Sept., (b) 02 UTC 18 Sept., (c) 05 UTC 18 Sept., (d) 07 UTC 18 Sept., and (e) 21 UTC 18 Sept. The gray line represents the R/V Atlantis ship track and the black cross on the ship track in panels a-e represent the location of the R/V Atlantis for the initialized back trajectory. The black rectangles around individual FLEXPART back trajectory intervals show the 2°x2° area in which C130 measurements were averaged and compared to downwind particle measurements (Table 3). The GOES-East satellite visible image for the 2°x2° area is included to present the cloud coverage corresponding to the time of the averaged 6-hour back trajectory interval. The GOES-East visible satellite image of the 2°x2° area around the ship initialization point in panel e is also included. Other trajectories were initialized at night and have no corresponding visible image.**



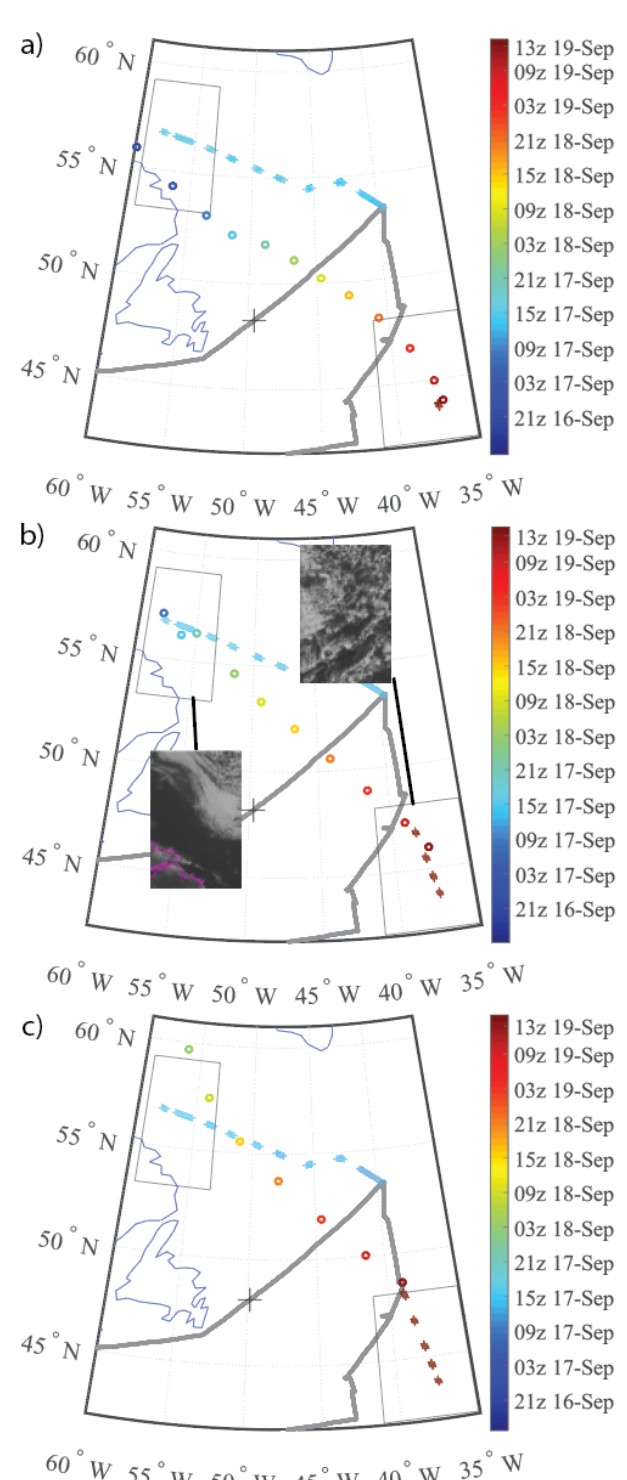


**Figure 7.** The location of FLEXPART back trajectories averaged over 6-hour intervals (shown as open circles) and low level (~0.1 km) C130 surface measurements (shown as colored lines) are both colored as a function of time. The colorbar range shifts by 40 minutes in each panel and is defined by the start and end of a 3-day FLEXPART back trajectory initiated at the C130 horizontal location at 500 m above the sea surface. The initialization times for the back trajectories are (a) 14:24 UTC 19 Sept., (b) 15:04 UTC 19 Sept., and (c) 15:44 UTC 19 Sept. The gray line represents the R/V Atlantis ship track and the black cross on the ship track in panels a-c represent the location of the R/V Atlantis for the initialized back trajectory. The black 5°x5° boxed area that overlaps the trajectory (in panel b) and FLT17 surface measurements identifies the C130 measurements used for comparison with downwind C130 measurements from FLT19 shown by the second 5°x5° area. GOES-East visible images are shown in panel b for both 5°x5° areas corresponding to the back trajectory initialization time and the 48-hour back trajectory interval that overlaps with FLT17 near-surface measurements. Table 3 contains comparisons between measurements made on FLT19 and measurements from the closed-cell and cloud-free region up-wind from FLT17.





80

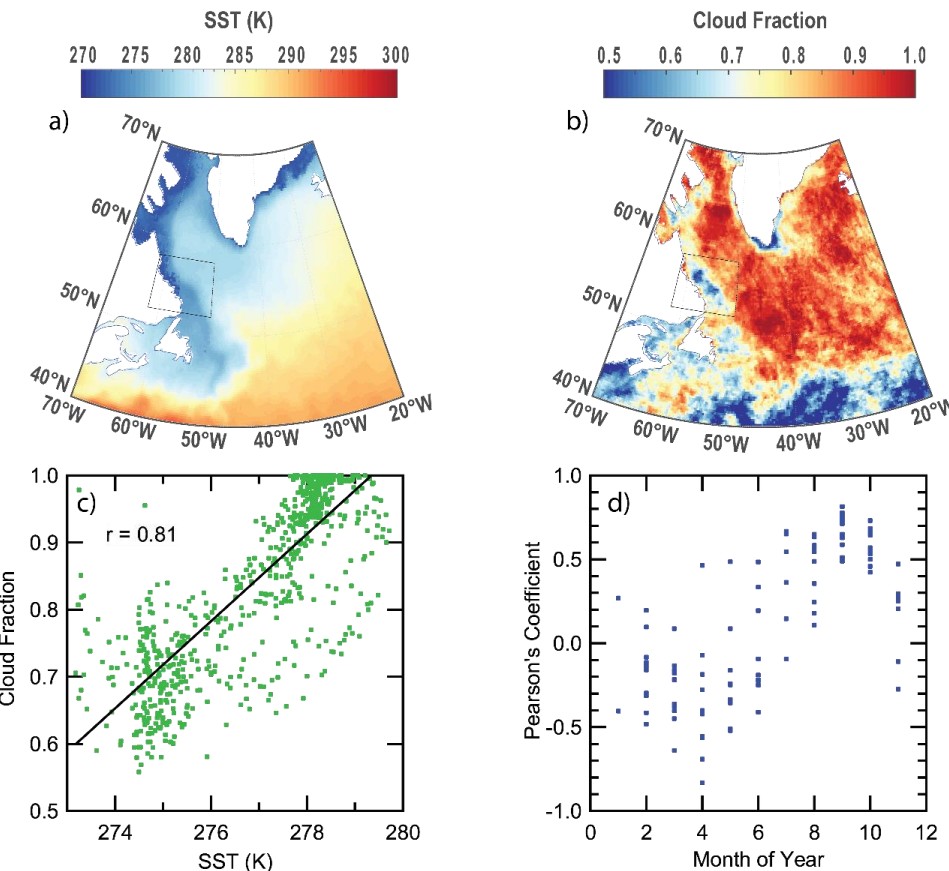

**Figure 8. September 2017 averaged satellite (a) SST and (b) CF. (c) The pixel-by-pixel comparison of SST and CF from the boxed area in panels a and b. (d) The Pearson's coefficient for the pixel-by-pixel linear regression comparing SST and CF for the boxed area (shown in panels a and b) for every month from Aug 2008-Jun 2019.**

85