# Peer review of "North Atlantic Ocean SST Gradient Driven Variations in Aerosol and Cloud Evolution along Lagrangian Cold-Air Outbreak Trajectories"

_Atmospheric Chemistry and Physics, 2021_

## Author Comment (AC1)

**REVIEWER 1**

Review of North Atlantic Ocean–Atmosphere Driven Variations in Aerosol Evolution along Lagrangian Cold-Air Outbreak Trajectories by K. Sanchez et al.

This manuscript presents an interesting case study which provides new insights on aerosol-cloud interactions in cold-air outbreaks. The data presented in this manuscript and the derived conclusions may be useful both for the modelling and experimental aerosol-cloud interactions research communities. The manuscript is well organized and clearly written. I recommend publications after the following (minor) comments are addressed.

We thank the reviewer for their feedback which has improved the readability of the figures and more clearly indicated the significance of our results. Please see direct responses to comments in blue text below.

Further considerations on the data robustness and uncertainty would be necessary in the manuscript, to allow a more sound interpretation of the results. If we take Table 3, which is a keystone of the manuscript, as an example, and consider the data displayed in Figure 4, we can conclude that not all the numbers reported in the Table are characterized by the same level of robustness. Clearly AMS concentrations are derived from few and quite scattered data, which make them more uncertain than other data presented contextually. I understand the scientific value of these data and the technical efforts necessary to obtain them and believe they are worth of publication. Nevertheless, at least, in the Table it should be reported the number of observations (n1, n2) used to derive the delta values so that the reader can judge about the robustness of the provided information. In alternative, the authors might evidence (e.g., with a *) which of the delta values are based on poorer statistics than the others, based on appropriate criteria.

We have added a supplemental table (Table S1) that includes the number of observations used in each comparison in Table 3.

The results reported in Table 3 are likely depending on the selection of the representative 10-minute periods of measurements used to calculate the deltas. Figure 2 suggests that a different choice, even by few minutes, may result in significantly different results. The authors should explain better how they selected the reference periods and show that their choice does not affect the results in a significant way (i.e., they should present the sensitivity of the results to the selection of the reference periods).

We agree that there were significant shifts in the measurements at times, shown in Figure 2. Upon implementing a statistical test on the Table 3 results (to indicate result significance, as the reviewer suggested in a later comment), we realized that the ship measurements represented in Figure 2 were mistakenly indicated as 10-minute averages. They were in fact 1-hour averages.

This information has been updated in the figure caption. Due to the significantly longer average and inclusion of the standard deviation in the result and the significance test (Z-score test), we believe that the choice of time does not significantly affect the results.

Specific comments

L270. I would invite the authors to indicate in brackets at what time the transition occurred, to help the reader in interpreting Figure 2.

Done.

L313-314. Please refer to my comments on Fig. 1 and Fig. 3.

Responses to comments on figures are below.

L388-390. A more robust statistic approach would make this assumption stronger. I invite the authors to apply a statistic test on the datasets to evidence which difference are statistically significant for a given confidence interval.

We thank the reviewer for this excellent suggestion. Assuming a confidence interval of >99%, 23 of the 28 comparisons in Table 3 were deemed significantly different. These comparisons that are statistically significant are now presented as bold print in the table.

L402. The authors may want to double check this sentence: "… resulted in the decreased the overall rate in aerosol particle".

The sentences has been corrected to read: "Either way, the different meteorological processes in the initially cloud-free and closed-cell region of the cold air outbreak likely resulted in the overall rate of particle removal between the two measurements relative to upwind trajectories that were clearly in the cold-air outbreak (Table 3)."

Table 1. Please correct the units of measurement in the caption (superscripts are missing). I have not clear the concept of "updraft-weighted updraft velocity", maybe some explanations are needed here..

Fixed and the following equations were added to the table footnote to explain the calculations:

$$[1]Updraft\ weighted\ CDNC = \frac{\sum_i CDNC_i W_i [w_i > 0\ m\ s^{-1}, CDNC_i > 2\ cm^{-3}]}{\sum_i W_i [w_i > 0\ m\ s^{-1}, CDNC_i > 2\ cm^{-3}]}$$

$$[2]Updraft\ weighed\ W = \frac{\sum_i W_i [w_i > 0\ m\ s^{-1}, CDNC_i > 2\ cm^{-3}]}{\sum_i [w_i > 0\ m\ s^{-1}, CDNC_i > 2\ cm^{-3}]}$$

Table 2. Please correct the units of measurement in the caption (superscripts are missing).

Fixed.

Figure 1. I believe that the manuscript would be more immediately comprehensible by a wider audience if more information were provided in Figure 1. I would invite the authors to mark the borders of the open cell, closed cell and clear sky regions object of their investigation on the satellite images of Figure 1. In alternative, they could provide the required information adding an extra Figure in the supporting information.

We have marked the borders of the relevant open-cell, closed-cell and clear sky regions in the bottom 2 panels of the figure. See below:

[Figure]

**Figure 1. GOES-East visible satellite imagery at two-hour intervals for 17 September 2017 (left column) and 19 September 2017 (right column). Purple lines outline the coast of eastern Canada and the southern tip of Greenland. Cyan and red lines represent the flight track at altitudes > 3km and < 3km, respectively, ±1 hour from the satellite image time (shown on the y-axis). The yellow line represents the ship position ±1 hour from the satellite image time. The gray line is the entire ship track for the NAAMES3 campaign. In panels e and f, the cloud-free, closed-cell, and open-cell region are highlighted with approximate outlines in green, white, and orange, respectively.**

Figure 3. The Figure could be improve by showing which data points refer to which of the considered regimes: closed cell, open cell, clear sky… This could be done by adding a horizontal bar at the bottom of the plot, marking the respective regimes.

We have added color coded horizontal bars to the bottom of the figure to differentiate closed-cell, open-cell, and cloud-free regions.

Figure 8. In panel d), it would be interesting to discriminate between statistically significant and not significant correlations (according to a chosen confidence interval). It can be done easily by using two different colours for significant and not significant R values data points.

Only correlations with $P < 0.05$ are included in this figure. We do not think it is necessary to included insignificant correlations so, to clarify, we have added the following text to the figure caption: "Correlations with $P > 0.05$ have been excluded."

According to the Journal guidelines, the Data availability statement should be separated from the Acknowledgement Section.

The acknowledgements section, financial support, and data availability sections are now separated.

---

## Author Comment (AC2)

**REVIEWER 2**

The manuscript presents a 3-days case study of a cold air outbreak. The study builds on combined observations from satellite, airborne and ship-based, as well as reanalysis data. The Lagrangian design of the observations provides insights into cloud morphology and its relations to particle concentrations and sea surface temperature. The sea surface temperature varies due to the Labrador current and controls the boundary stability, thus clouds. The study shows that although air masses origin for both trajectories had similar aerosol characteristics, the aerosol characteristics downwind were different, attributed to aerosol-clouds interactions along the trajectory. Linking these findings to the role SST in the two different trajectories is interesting. Although the study focuses on a single case study, the authors show that the observed pattern they identified occurs annually due to the location of ocean currents in the region, enhancing the relevance of the study.

The manuscript is clearly written and I would recommend publications after addressing the following issues:

We greatly appreciate the reviewer's feedback and believe that overall, their comments and suggestions have improved the quality of the manuscript. Please see direct responses to comments in blue text below.

Major comments:

1. In the introduction the authors nicely describe in details the theory of Sc-to-Cu transitions that was developed based on subtropical Stratocumulus cloud transitions. I am not sure the same arguments apply also for cold air outbreaks. I would like the authors to discuss and show how the decoupling and breakup processes in the stratocumulus regions are relevant to cold air outbreak as well (e.g., SST gradients, subsidence rates, humidity above the inversion layer and boundary layer deepening are fundamental ingredients of the Sc-to-Cu transitions. Are those important for cold air outbreak as well? Or that ocean fluxes due to the large air-sea temperature is the main cause of the transition? See, e.g., https://doi.org/10.1002/2017JD027031).

From the literature, it is not well known if the theories on Sc-to-Cu transitions in the subtropics also equally apply to Sc-to-Cu transitions in cold air outbreaks, however it is not even clear which of the theorized processes that influence Sc-to-Cu cloud transitions in the subtropics are significant. Differentiating the relative impact of each process has proven difficult since all the processes tend to occur together. The manuscript that the reviewer has pointed to does suggest the advection of cold air over warm water is possibly a large driving factor, but this process leads to many of the other processes mentioned (boundary layer deepening, entrainment, enhanced precipitation, boundary layer decoupling, etc.). For this reason, we have added the following sentence to the introduction:

"While the advection, entrainment, and microphysical processes described all play a role in the occurrence of a closed-cell to open-cell transition, recent evidence suggests the advection of cool air over warmer waters is the initial driver of this transition, as indicated by the correlation of this event with the surface forcing and static instability of the boundary layer (Mccoy et al., 2017)."

Also, we believe we have directly discussed how decoupling and the break-up processes in stratocumulus regions are relevant to the Sc-to-Cu transition, though we could have better linked it to transitions occurring in cold-air outbreaks. For clarity, we have updated the following paragraph from the introduction with the highlighted text:

"Cold-air outbreaks commonly occur in the North Atlantic and are characterized by an expansive area of closed-cell stratocumulus clouds that transition to form open-cell convective clouds. Cold-air outbreaks are of recent scientific interest because they provide well-posed cases to study regime-dependent cloud radiative properties and to improve model representation of the cloud evolution across these regimes (Field et al., 2014; Fletcher et al., 2016; Tselioudis et al., 2013). Such transitions have been extensively studied with modeling and observations in the subtropics, where the break-up of the closed-cell region is reportedly driven by the decoupling of the cloud layer from the surface (i.e., when an inversion separates the boundary layer into two distinct layers, impeding mixing between the layers) (Abel et al., 2017; Albrecht et al., 2016, 1995; Berner et al., 2013; Bretherton and Wyant, 1997; Christensen et al., 2020; Ghate et al., 2015; Lloyd et al., 2018; de Roode et al., 2016; Sandu and Stevens, 2011; Wood et al., 2011, 2017; Yamaguchi et al., 2017). While less studied, the transition from closed-cell to open-cell convective clouds have been observed in marine cold-air outbreaks due to similar driving mechanisms (Abel et al, 2017; Fletcher et al., 2016). There are two main processes that lead to the decoupling and stratocumulus-to-cumulus transition. First, advection of cold Arctic air over warmer sea surface temperatures (SST) increases sensible and latent heating, causing the marine boundary layer depth to increase. With increased boundary layer depths and surface heating, the top-down circulation of marine stratocumulus clouds (driven by cloud top radiative cooling and cloud top evaporative cooling), can no longer extend to the ocean surface, initiating the decoupling of the marine boundary layer (Albrecht et al., 1995; de Roode et al., 2016; Wang and Feingold, 2009). Second, cloud-top entrainment of free tropospheric air warms and dries the decoupled cloudy layer as the below-cloud surface-coupled layer continues to moisten from the latent heating, thereby strengthening the inversion between the two layers (Albrecht et al., 1995; Bates et al., 1998a; Bretherton and Wyant, 1997; Zhou et al., 2015). The progression of these two processes decreases the altitude of the lifting condensation level of rising air and increases the altitude of the lifting condensation level of sinking air, respectively, causing the stratocumulus cloud base to rise until it has completely evaporated, while also forming convectively driven cumulus clouds with lower cloud bases. The progression rate of the stratocumulus-to-cumulus transition is heavily dependent on the strength of the decoupling in the MBL and how fast the decoupled cloud layer dries through cloud-top entrainment and precipitation. Both cloud-top entrainment rates and precipitation rates are heavily influenced by the cloud layer aerosol properties (Abel et al., 2017; Ackerman et al., 2004; Albrecht, 1989; Berner et al., 2013; Hill et al., 2009; Jiang et al., 2006; Lu and Seinfeld, 2005; Stevens et al., 2005; Twomey, 1977; Yamaguchi et al., 2017)."

2. The authors emphasis the role of meteorology. As far as I understand, by meteorology the authors mean the SST (and subsequently stability) along the air mass trajectories. So why not calling it stability? Meteorology in that sense is somewhat miss leading, since meteorology is

more than SST and stabilty. I would ask the authors to explain what they mean by meteorology. The explanation given in Line 161 for the different meteorological condition is not informative with respect to the role meteorology.

We believe the meteorology is worth emphasizing. While SST and resulting stability may have initiated observed differences in the meteorology (the cloud droplet concentration, precipitation, cloud fraction and resulting cloud radiative forcing), the differences and evolution of the meteorology along the trajectory and Sc-to-Cu transition (and the difficulty in forecasting such transitions) are why the results are of interest. We also believe we are clear at indicating that the SST driven stability played a major role in the resulting meteorological differences in both section 3.2 'C130 Vertical Profile Stability' and section 3.4 'Ocean SST Control of Atmospheric Stability.' We have updated the following sentence in the conclusion section to more clearly emphasize the role of SST-driven atmospheric stability in the observations.

> "While particle number concentrations and mass composition are similar within the northwest (i.e., upwind) portion of the study region, the cloud conditions are very different, with a notable clear air region above the Labrador Current driven by cool sea surface temperatures stabilizing the marine boundary layer and a stratiform cloud and less stable region immediately to the northeast over warmer waters."

3. The case study presents a trajectory that passed through a cold air outbreak. But it is the different is SST between that trajtory and another one that makes it an interesting story. I think that emphaising this in the title, rather than the cold air outbreak, would be more infomative.

We agree that the SST difference is what makes the story interesting and originally incorporated that with the text 'Ocean-Atmosphere Driven Variations' in our title. We have changed the title to the following to more clearly emphasize the SST as a major part of the findings, but still emphasize the cold-air outbreak as well due to the importance of these measurements.

Previous title: "North Atlantic Ocean–Atmosphere Driven Variations in Aerosol Evolution along Lagrangian Cold-Air Outbreak Trajectories"

New title: "North Atlantic Ocean SST Gradient Driven Variations in Aerosol and Cloud Evolution along Lagrangian Cold-Air Outbreak Trajectories"

Specific comments:

Line 72: Not clear what the authors mean by "particle trajectory".

We have changed the text to "FLEXible PARTicle (FLEXPART) Lagrangian trajectories"

Line 114: Instead of "break-up" I suggest writing "dissipation", as the clouds are drying, as the authors write.

The text has been updated.

Line 243: Please elaborate on "cloud conditions".

We have changed 'cloud conditions' to 'cloud coverage' to be more specific.

Line 271: https://doi.org/10.1002/2015JD023176 and https://doi.org/10.1073/pnas.261712099 showed case studies of continental air associated with overcast cloud regime over the north east Atlantic. Might be relevant.

The FLEXPART back trajectories and low radon ($< 500$ mBq m$^{-3}$), CO, and BC concentrations (shown in supplemental figures S3 and S4) all suggest the aerosol in these cases were likely of clean marine origin and that neither continental nor anthropogenic aerosols had significant contributions. Also, the trajectories originate from the northwest Atlantic (Figures 6, 7, S4), while the referenced publications indicate the continental air associated with the overcast cloud regime is from the northeast Atlantic/Europe.

Line 325: What do you mean by "differences in meteorology"?

We have changed 'differences in meteorology' to 'differences in cloud coverage' to be more specific.

Line 353-356: How decoupling can form along with a rapid increase in SST? Wouldn't it enhance cumulus formation from the surface? Can you shows SST along the trajectory?

While counterintuitive, the mechanisms behind the formation of the decoupled layer in the marine boundary layer with rapidly increasing SST are explained in the introduction (copied below). The SST is shown along the trajectory with points at the surface (0 km) on Figure 5.

"Cold-air outbreaks commonly occur in the North Atlantic and are characterized by an expansive area of closed-cell stratocumulus clouds that transition to form open-cell convective clouds. Cold-air outbreaks are of recent scientific interest because they provide well-posed cases to study regime-dependent cloud radiative properties and to improve model representation of the cloud evolution across these regimes (Field et al., 2014; Fletcher et al., 2016; Tselioudis et al., 2013). Such transitions have been extensively studied with modeling and observations in the subtropics, where the break-up of the closed-cell region is reportedly driven by the decoupling of the cloud layer from the surface (i.e., when an inversion separates the boundary layer into two distinct layers, impeding mixing between the layers) (Abel et al., 2017; Albrecht et al., 2016, 1995; Berner et al., 2013; Bretherton and Wyant, 1997; Christensen et al., 2020; Ghate et al., 2015; Lloyd et al., 2018; de Roode et al., 2016; Sandu and Stevens, 2011; Wood et al., 2011, 2017; Yamaguchi et al., 2017). While less studied, the transition from closed-cell to open-cell convective clouds have been observed in marine cold-air outbreaks due to similar driving mechanisms (Abel et al, 2017; Fletcher et al., 2016). There are two main processes that lead to the decoupling and stratocumulus-to-cumulus transition. First, advection of cold Arctic air over warmer sea surface temperatures (SST) increases sensible and latent heating, causing the marine

boundary layer depth to increase. With increased boundary layer depths and surface heating, the top-down circulation of marine stratocumulus clouds (driven by cloud top radiative cooling and cloud top evaporative cooling), can no longer extend to the ocean surface, initiating the decoupling of the marine boundary layer (Albrecht et al., 1995; de Roode et al., 2016; Wang and Feingold, 2009). Second, cloud-top entrainment of free tropospheric air warms and dries the decoupled cloudy layer as the below-cloud surface-coupled layer continues to moisten from the latent heating, thereby strengthening the inversion between the two layers (Albrecht et al., 1995; Bates et al., 1998a; Bretherton and Wyant, 1997; Zhou et al., 2015). The progression of these two processes decreases the altitude of the lifting condensation level of rising air and increases the altitude of the lifting condensation level of sinking air, respectively, causing the stratocumulus cloud base to rise until it has completely evaporated, while also forming convectively driven cumulus clouds with lower cloud bases. The progression rate of the stratocumulus-to-cumulus transition is heavily dependent on the strength of the decoupling in the MBL and how fast the decoupled cloud layer dries through cloud-top entrainment and precipitation. Both cloud-top entrainment rates and precipitation rates are heavily influenced by the cloud layer aerosol properties (Abel et al., 2017; Ackerman et al., 2004; Albrecht, 1989; Berner et al., 2013; Hill et al., 2009; Jiang et al., 2006; Lu and Seinfeld, 2005; Stevens et al., 2005; Twomey, 1977; Yamaguchi et al., 2017)."

In addition, the authors relate closed-to-open cells transition to decoupling. The decoupling is part of the Stratocumulus break up to cumulus clouds, while closed to open cells transition process is more associated to drizzle formation (e.g., https://doi.org/10.5194/acp-6-2503-2006).

We are confused on what the reviewer believes the difference is between "stratocumulus break up to cumulus clouds" and "closed to open cell transition". We believe they are synonyms. In the introduction, we provided background on the fact that the closed to open cell transition is influenced by decoupling and drizzle formation. For a broader overview, we would refer the reviewer back to the reference they provided earlier (https://doi.org/10.1002/2017JD027031, McCoy et al., 2017).

How is the decoupling measured?

Decoupling is not directly measured but is often based on the potential temperature or virtual potential temperature change with altitude in the boundary layer. Here the potential temperature is used (Figure 5) since it is directly comparable to the SST and there is not a significant difference in the potential temperature slope with altitude and virtual temperature slope with altitude. In a fully coupled boundary layer, the marine boundary layer potential temperature would be approximately constant with height outside of the cloud, then rapidly increase with height at the boundary layer inversion. A decoupled boundary layer would have a layer in between with a gradient in potential temperature greater than that of the coupled surface layer gradient and less than the inversion layer gradient.

Have you looked on the relationship between decoupling and warm advection? (e.g., https://doi.org/10.1029/2018GL078122 and DOI:10.1002/essoar.10507144.1)

Yes we have addressed the warm air advection that occurs in this specific case with the following statement: "The cause of the differing atmospheric stability in this region is the low SST (Figure 5a), which cools and stabilizes the marine boundary layer, preventing the vertical transport of water vapor and cloud formation."

Line 412: check for typos.

Fixed.

Line 499: Can entertainment of aerosol from the free troposphere explain aerosol properties (e.g. https://doi.org/10.1175/BAMS-D-13-00180.1)

The possible entrainment of particles is certainly something that should be addressed. As discussed, part of the stratocumulus to cumulus transition processes involves entrainment of warm/dry free tropospheric air which tends to dry and decouple the marine boundary layer. At the same time, particles could also be entrained into the marine boundary layer. However, once decoupled, the layer would act as a buffer between the free troposphere and ocean surface, minimizing the transfer of particles in between them. The marine boundary layer was not always decoupled, as indicated by the potential temperature profiles in Figure 5, therefore, during these stages entrainment of particles is more likely to occur. The new supplemental figure below shows CN concentration profiles using the same profiles as in Figure 5. Based on a boundary layer inversion height of 1-1.5 km (range derived from Figure 5), the lower free troposphere particle concentrations are sometimes greater than the surface observed concentration and therefore could act as a net source of CCN, particularly in the eastern portion of the 19 September case where surface particle concentrations are elevated to the west and profiles are less stable to the west. We have updated our statement to the following:

"For both cases, the change in the organic to sulfate mass ratio is unexpected as sulfate components are more hygroscopic and therefore, more likely to activate to form cloud-droplets and be removed through precipitation. Vertical profiles of CN suggest the free troposphere was a source of sulfate entrained into the marine boundary layer, especially for the case with less stable boundary layers, which would enhance vertical entrainment. It is also possible that the sulfate resided in smaller particles or more of the sulfate was removed through precipitation and subsequently more sulfate particles were formed via gas phase or in-cloud oxidation of $SO_2$."

The following text was also added to section 3.3.2.

In reference to the stratocumulus to cumulus trajectory:

"Vertical profiles of CN (Figure S5a), corresponding to the potential temperature profiles in Figure 5, mostly show lower particle concentrations above 1-1.5 km (the range of boundary layer heights based on Figure 5) relative to the surface, suggesting it is likely not a net source of particles in this case. Also, the decoupling of the boundary layer acts as a buffer, inhibiting the transfer of particles and gaseous precursors between the surface and lower free troposphere."

In reference to the cloud-free to cumulus trajectory:

"The upwind vertical profile of $CN_{>10nm}$ on FLT17 in Figure S5a shows there was a strong negative gradient with altitude in the cloud-free region (at ~58°W), suggesting the decrease in the observed surface CN>10nm between FLT17 and FLT19 is likely due to dilution when mixing with lower concentrations higher in the boundary layer upon destabilization of the boundary layer. Also, the reduction in stability likely enhanced entrainment with the lower free troposphere, which is shown to have relatively high particle concentrations in the 19 September case (Figure S5b). The formation of sulfate particles due to mixing between the marine boundary layer and lower free troposphere is supported by another NAAMES study (Sanchez et al., 2018) and others (Clarke et al., 1999; Dzepina et al., 2015; Moore et al., 2003) and likely explains at least part of the increase in sulfate particle mass along this trajectory."

[Figure]

**Supplemental Figure S5. Vertical profiles of the CN>10nm concentration from C130 flights that occurred on (a) 17 September 2019 and (b) 19 September 2019 (Figures 3a and 4a). The same vertical profiles are used to show potential temperature in Figure 5. The point color represents the longitude at which the measurement was made. Air mass transitions from opened-cell to closed cell clouds and from closed-cell to cloud-free air on the 17 September**

**2019 flight occurred at approximately 52 °W and 56 °W, respectively. None of the vertical profiles are in the closed-cell region for the 19 September 2019 flight. In-cloud measurements are excluded and a 30 second running averages is applied to improve readability.**

Line 506: Do you have an estimate of the replenishing rate?

The reviewer is referring to the following text "Given the localized area of the higher ocean chlorophyll and biological activity to the northwest (Figure S8), we speculate that while both organic and sulfate aerosol species are rapidly depleted by wet scavenging along the cold-air outbreak trajectory (i.e., the closed-to-open-cell transition), the sustained secondary sulfate source is able to replenish the sulfate aerosol concentration."

Unfortunately, we do not have an estimate of the replenishing rate. The replenishing rate is dependent on complex chemistry that is not completely understood and is also dependent upon the concentration of precursors emitted from the ocean that are also not fully understood. Alternatively, one would need to know the removal rate to calculate the replenishing rate.

Figure 1: Can resolution be improved? Consider also enhancing the colors in the darker images to improve visibility.

The resolution is now increased and the top panels have been brightened slightly to improve visibility. (See figure below)

September 17 September 19

[Figure]